# `RP-Mod` & `RP-Crowd`: Moderator- and Crowd-Annotated German News Comment Datasets

**Dennis Assenmacher**[*]
University of Münster
GESIS

**Marco Niemann**
University of Münster

**Kilian Müller**
University of Münster

**Moritz V. Seiler**
University of Münster

**Dennis M. Riehle**
University of
Koblenz-Landau

**Heike Trautmann**
University of Münster
University of Twente

## Abstract

Abuse and hate are penetrating social media and many comment sections of news media companies. These platform providers invest considerable efforts to moderate user-generated contributions to prevent losing readers who get appalled by inappropriate texts. This is further enforced by legislative actions, which make non-clearance of these comments a punishable action. While (semi-)automated solutions using Natural Language Processing and advanced Machine Learning techniques are getting increasingly sophisticated, the domain of abusive language detection still struggles as large non-English and well-curated datasets are scarce or not publicly available. With this work, we publish and analyse the largest annotated German abusive language comment datasets to date. In contrast to existing datasets, we achieve a high labelling standard by conducting a thorough crowd-based annotation study that complements professional moderators' decisions, which are also included in the dataset. We compare and cross-evaluate the performance of baseline algorithms and state-of-the-art transformer-based language models, which are fine-tuned on our datasets and an existing alternative, showing the usefulness for the community.

## 1  Introduction

Abusive language and hate speech are almost as old as modern society [cf., 62, 45]; however, they are massively fuelled through internet media communication in recent years [49, 56]. Some even consider abusive language a major threat to free online discourse [49]. While social media platforms are taking the major hit [49], newspapers are severely affected as well [14, 40]. Many news outlets are caught between the willingness to provide a participatory discourse on the one side and the increasing legal regulation and the comment moderation economics on the other side [5, 47].

As an alternative to closing down comment sections, (semi-)automated moderation solutions using machine learning (ML) and natural language processing (NLP) have become hot topics in research, and practice [49, 56]. One major impediment is the scarcity of published, large-scale comment datasets that can serve as the basis for model training and optimisation [46, 57]. For all non-English and non-Social Media outlets, the data situation is even direr [57]. This holds explicitly for datasets from German news publishers, with the `DerStandard` dataset being the only publicly available training data [52]. However, with $\sim 3,600$ labelled entries and four years of age, there is a plausible need for an updated and extended German news comment dataset.

---

[*]Corresponding author. E-Mail: `dennis.assenmacher@uni-muenster.de`

35th Conference on Neural Information Processing Systems (NeurIPS 2021) Track on Datasets and Benchmarks.

This paper describes and publishes the most extensive annotated German (news) comment data set to date. We collaborate with a major German newspaper to have full access to moderated and unmoderated comments. Seeking to cross-validate community managers' decision-making with public perception and to provide more fine-grained labelling, the comments were also labelled through the crowdworking platform Crowd Guru[2]. To provide guidance on the opportunities linked to this dataset, we present an initial descriptive analysis and conduct a set of experiments to investigate and show the suitability of the annotation data to detect abusive content in a supervised setting. Moreover, we cross-evaluate trained models on a different German comment dataset to get insights into our models' generalisability.

## 2 Related Work

Detecting hate speech, respectively abusive language, in online comments is one of the classic examples of supervised learning. This directly implies that this domain massively depends on labelled datasets to advance the state-of-the-art classification models. Several comprehensive surveys were published throughout the last years, giving an overview of the current dataset landscape [21, 34, 46, 57]. Common patterns across all studies are English being the predominant language and Twitter being the most prevalent data origin. Vidgen and Derczynski [57] attribute this to the conveniently usable Twitter API and caution against the over-use of Twitter data. This has a semantic and a syntactic component, as Tweets are non-representative of the public opinion [35], and through Twitter's length restrictions, length and style typically vastly differ from other outlets [4, 57]. Furthermore, Poletto et al. [46] point out that many reported and used datasets are not publicly available.

Datasets originating from comments on the webpages of news outlets are rare, representing six [46] to seven [57] out of roughly 60 commonly referenced ones, while the others usually originate from social media. All datasets that could be identified from [21, 34, 46, 57] have been aggregated in Table 1 complemented by items found throughout our review efforts. The list holds no claim of completeness, but being an aggregate already indicates the sparsity of news-related comment datasets. However, as Vidgen and Derczynski [57] suggest, the legally safe publication of such data is complex, as it requires official approval of the hosting platform and often even legal changes on their side (e.g. changes in privacy policies).

Table 1: Publicly Available Abusive Language Datasets from News Outlets

| Lang. | Ref. | Year | Outlet | Total | Prob. | Unprob. |
|---|---|---|---|---|---|---|
| Arabic | [39] | 2017 | AlJazeera | 32,000 | 6,080 | 25,920 |
| Croatian | [32] | 2018 | 24sata | 17,042,965 | 310,147 | 16,732,818 |
| | [54][3] | 2020 | 24sata | 21,548,192 | 454,057 | 21,094,135 |
| | [54] | 2020 | Vecernji | 9,646,634 | 447,290 | 9,199,344 |
| Czech | [55] | 2017 | Mixed | 1,812 | 702 | 1,110 |
| English | [22] | 2017 | Fox News | 1,528 | 435 | 1,093 |
| | [55] | 2017 | Mixed | 1,007 | 250 | 757 |
| | [30] | 2019 | Globe and Mail | 1,121 | 219 | 902 |
| Estonian | [54] | 2020 | Ekspress | 31,487,359 | 2,813,186 | 28,674,173 |
| French | [55] | 2017 | Mixed | 487 | 144 | 343 |
| German | [52] | 2017 | DerStandard | 3,599 | 303/ 282 | 3,296/ 3,317 |
| | [55] | 2017 | Mixed | 1,112 | 149 | 963 |
| Greek | [44][4] | 2017 | Gazetta | 1,449,600 | 489,222 | 960,378 |
| Italian | [55] | 2017 | Mixed | 649 | 249 | 400 |
| Korean | [37] | 2020 | Korea Herald | 9,341 | 3,905 | 5,436 |
| Portugese | [17] | 2017 | g1.globo | 1,250 | 419 | 831 |
| Slovene | [32] | 2018 | MMC RTV | 7,569,686 | 630,961 | 6,965,725 |

Several of the omnipresent patterns in the above-stated reviews do not apply to the news comment datasets. For example, data for the otherwise common English language is scarce as the datasets are few and small ($< 2,000$ items each). Here, the unavailability turns out to be problematic again, as larger datasets exist [42] but are not made available, respectively, are not yet available [2].

---

[2] https://www.crowdguru.de
[3] Presumably an extension of the dataset of [32].
[4] The smaller subsets of the main dataset discussed in [44] are not further considered.

Contrasting to this, three small East European countries (Croatia, Estonia, and Slovenia) are the origin of the by far most extensive news media comment datasets (all $> 7.5$ million items). Given the sparsity of datasets per country, any insights regarding the diversity of comment sources are complicated; however, there seem to be no media partners, who are exceptionally easy to cooperate with, that would match the popularity of large Social Media platforms such as Twitter. Similar to the general situation for abusive language comment datasets, the news media datasets have varying annotation schemes, typically tailored to each use case. To provide a notion of the distribution of problematic and unproblematic content, we transferred the multi-class schemes into a simple binary mapping, which exhibits a common trait: A class imbalance towards unproblematic comments [57]. Regarding the assignment of labels, the majority of the datasets have been labelled by crowdworkers, and experts [22, 30, 37, 39, 17, 55], very similar to most Social Media datasets. However, the most extensive datasets (all $> 1$ million items) were labelled by the community managers, and journalists of the outlet [32, 44, 54]; the comments by [44] were partially even relabelled subsequently. Hence, there is little insight into whether community managers and the community have similar perceptions on abusiveness—with the study [60] indicating the opposite. This underlines two of the big opportunities and problems of news media comment datasets: For larger outlets, many will come with moderation decisions; however, they must also be willing to share these with the public. Overall, such organisations' willingness to share seems comparatively low, as most datasets are from 2017 and 2018, with no upward trend for 2020.

## 3    Dataset Creation

A more general pitfall for creating datasets is a lack of a comprehensive description of its creation (e.g., sources, label schema, annotators, . . . ) [3]. Adverse consequences are far-reaching, from sub-par performance to the introduction of biases to missing reproducibility [3, 23]. As our work touches on critical ethical concerns such as the freedom of speech, we strive to avoid this fallacy by providing extensive insights into the dataset characteristics [3, 23].

### 3.1    Data Acquisition

The presented data were acquired through collaboration with one of Germany's major newspapers, the `Rheinische Post (RP)`[5] in the course of a research project. The comments were posted on our partner's website between November 2018 and June 2020, where registered users are free to comment on most published articles. Each incoming comment is pre-moderated by one of the community managers of the RP. In addition to the raw comments, we receive the moderators' decision (called `RP-Mod` dataset) in form of a binary coding (non-abusive [0] vs. abusive[6] [1]).

The privacy policy statement of the RP's comment section was adjusted to allow for a research-oriented use of the comment data. All users whose comments we received were aware of their data being used for research purposes and consented to it. This circumvents several ethical and legal problems otherwise present with scraped or manually extracted data (copyrights, privacy, . . . ).

### 3.2    Labelling Schema

The thoughtful composition of the labelling schema is paramount, as the labelled data represents the model performance baseline [19]. Aside from the general complexity of annotation, abusive language labelling suffers from the required consideration of legal constraints (which need interpretation and vary nationally) [8, 56] and linguistic intricacies such as sarcasm and irony, which are structurally similar to abusive language [21]. Hence, pragmatic and abstract labels [57], as well as the assignment of multiple labels to an individual comment [27], are recommended.

Most papers so far either reused the annotation schema of the data-providing organisation [52, 54] or of a prior publication [22, 32, 39, 44]. To satisfy the above-stated criteria and to use a more integrative schema, we selected the approach of [41] for the annotation of our corpus. It provides a theory-deduced labelling schema (considering linguistics, academia, and law), comes pre-configured for the German language setting, encourages multi-label annotation, and includes organisation-specific labels.

---

[5]`https://rp-online.de`

[6]Please note that in the context of this paper and the dataset we will use the terms "abusive" (label name) and "reject(ed/ion)" (consequence of a comment being abusive) synonymously.

Table 2: Full Labelling Schema translated from and based on [41]

| | Label | Explanation |
|---|---|---|
| **Theory-Deduced** | sexism | Attacks on people based on their gender (identity), often with a focus on women. |
| | racism | Attacks on people based on their origin, ethnicity, nation (typ. meant to incite hatred). |
| | threats | Announcements of the violation of the physical integrity of the victim. |
| | insults | Denigrating, insolvent or contemptuous statements (left without further specification). |
| | profane language | Usage of sexually explicit and inappropriate language. |
| **Orga.-Specific** | meta / organisational | Organisational content such as request on why specific posts or commenters have been blocked. Not abusive per se, but heat up conversations providing little to no contribution. |
| | advertisement | Comments advertising unrelated services or products or linking to such. Not abusive per se; but lead to a deterioration of discourse sentiment and quality. |

We followed the advice of closely collaborating with practitioners [28] and scheduled a workshop with the community managers of the RP. It validated the theory-deduced schema of [41] and led to the inclusion of two additional, organisation-specific labels to account for problematic comments not yet accounted for by the theory-deduced schema (for the full labelling schema cf. Table 2).

### 3.3 Comment Annotation Study

To apply the novel annotation schema and to cross-check public perception with the moderators' assessments, we conducted a large-scale crowdworking study with Crowd Guru as the service provider[7]. This is not only the de-facto standard for annotating large-scale datasets [59], but in our case, we also profit from crowdworkers being laymen and not linguistic experts [20, 59].

To balance the number of comments labelled and capturing a diverse/representative feedback per comment, we decided to obtain five annotations per comment and at max. 1,000 votes from an individual crowdworker. In combination with extant budget restrictions and pre-study findings indicating an average comment could be labelled in 18 seconds, we determined the quantity of annotatable comments to be 85,000 (paying German minimum wage to all annotators). Like most prior annotation studies, this implied sub-sampling our originally larger dataset (cf., e.g., [30]). As some of our newspaper comments exhibit lengths of up to $25,000$ characters, we first decided to restrain comments submitted for crowd annotation to 500 characters—an actionable number according to our pilot study. This is in line with recommendations to keep annotation tasks short to avoid quality drops, or subpar participation [7, 59]. Since the majority of the available datasets suffer from an insufficient amount of problematic comments [33], the decision was made to oversample them intentionally. Hence, all $7,141$ comments blocked by the community managers were chosen. The remaining $77,859$ comments were randomly sampled from all unblocked comments.

For quality assurance, all annotators were required to be (close to) native German speakers. Furthermore, we implemented multiple quality checks, including so-called attention checks, gold standard comments, and participant pre-screenings [25, 53]. We used 56 gold standard comments, which we sourced from a pre-study [9], where these comments have been labelled uniformly. Furthermore, we provided 20 attention check comments, which explicitly asked the crowdworkers to select a specific answer [25, 43].

To participate in the crowdstudy, each interested crowdworker[8] had to go through a briefing first, where they were informed about the subject, the annotation task, and the risk of exposure to abusive comments (cf., Appendix A.1.1). Afterwards, each crowdworker had to complete a pre-screening consisting of 20 gold standard comments, of which 15 had to be identified correctly to be eligible for the study. Once admitted to the study, each crowdworker was allowed to label up to 1,000 comments, interleaved with an attention check every 20 comments, and a gold-standard comment every 50 comments. For each comment the crowdworker has to decide first, whether the comment is publishable. Whenever a decision is made against publishing it, one or multiple labels from our annotation schema in Table 2 had to be assigned (otherwise the selection is not visible). Enabling multi-label annotation follows the recommendations of [27] and [41], who argue that text items

---

[7]The selection was conducted under adherence to German procurement laws.

[8]Crowd Guru requires all their crowdworkers to be at least 18 years of age (legal age of maturity in Germany). Hence, no minors were involved in the annotation process.

Table 3: Dataset Characteristics

| Dataset Characteristic | Value |
|---|---|
| Number of comments | 85,000 |
| Number of (crowd)-annotators per comment | 5 |
| Number of articles | 22,499 |
| Number of users | 5,235 |

| | Metric | | | |
|---|---|---|---|---|
| Dataset Characteristic | Min | Max | Mean | Std |
| Comments per topic | 1 | 80 | 3.78 | 5.52 |
| Number of posts per user | 1 | 1666 | 16.24 | 71.68 |
| Characters per comment | 0 | 500 | 227.13 | 127.65 |

(e.g., comments) can be part of multiple classes in parallel. An overview of the complete annotation process, as well as mock-ups of the annotation interface, can be inspected in the Appendix.

The successful annotation of the crowdworkers resulted in the second dataset, `RP-Crowd`. The full dataset encompasses the moderator annotated `RP-Mod` and the crowdworker annotated `RP-Crowd`.

## 4 Dataset Characteristics

Base statistics of the dataset[9] are depicted in Table 3. The information contained in the dataset is split into twelve features as described in Table 4. Each entry in our dataset contains a numeric ID, as well as the text of the comment (the raw text is provided). Additionally, the moderation decision of the Rheinische Post, which is a binary label, is included (`RP-Mod`). The remaining features (together with the "ID" and "Text") constitute the `RP-Crowd` dataset. For each comment and each label (cf., Table 2), we summed up the votes of all five voting crowdworkers, so that each comment can have 0-5 rejections per label. This is complemented by the "Rejection Count Crowd" column, which contains the sum of initial reject decisions (cf., Section 3.3) per comment and the "Reject Crowd" column, which is set to "1", whenever three crowdworkers rejected a comment (simple majority vote).

Within Figure 1, demographic indicators of the worker population can be inspected. Male and female workers are almost evenly distributed (300 and 301 respectively), whereas two people identified as non-binary. Unsurprisingly, most of the participants originally come from Germany (482) and Austria (43), while the remaining workers (78) originate from various countries from all over the world. This is as expected since the study explicitly targeted a German-speaking audience.

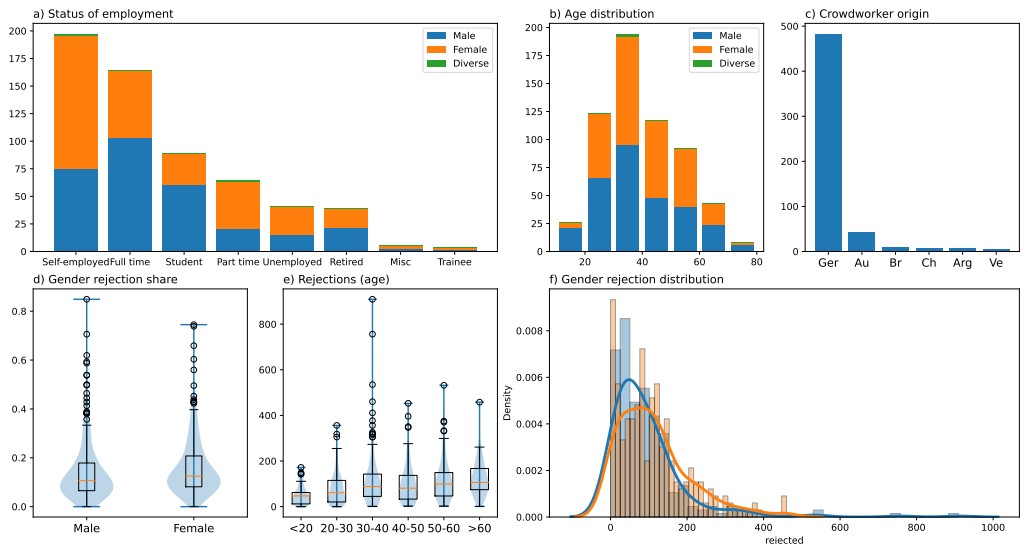

Figure 1: Demographic overview of the Crowdworkers

The majority of the workers are currently employed (425) or studying (89). A smaller proportion of the participants are either unemployed (41) or already retired (39). This fits with the age distribution, where 75% of the users are between 20 and 60 (64 being the average retirement age).

---

[9]The dataset is available on `zenodo` via `https://doi.org/10.5281/zenodo.5242915`.

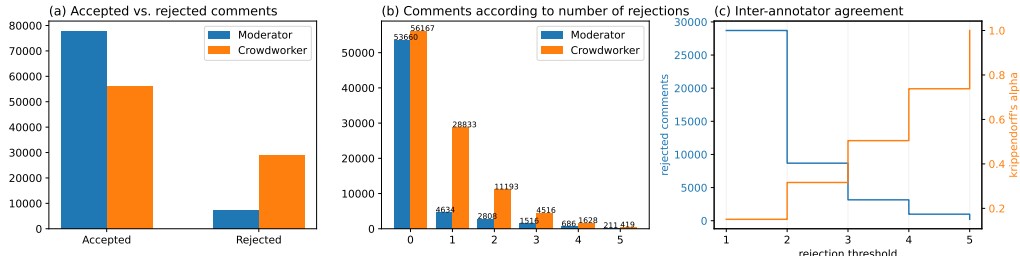

Figure 2: Comparison Moderator vs. Crowdworker

In line with the results of recent studies [1], the rejection tendency of males and females does not seem to differ from each other significantly. While the most extreme outlier, with a relative rejection rate of 0.85 (85% of the comments classified by the worker are annotated as abusive), belongs to the males, the females' median rejection rate is slightly higher (0.1 vs. 0.08). Figure 1 (d) shows the relative rejection for each gender. Both genders exhibit a similar low rejection rate, which is not surprising due to the moderators' large number of accepted posts. As depicted in Figure 1 (e), users tend to reject comments more frequently with increasing age.

A more in-depth investigation of the difference between acceptance and rejection decisions of moderators and crowdworkers surprisingly shows that the latter tend to reject comments more rigorously (see Figure 2 (a)). Figure 2 (b) shows the acceptance/rejection agreement between the crowdworkers, coloured in orange, and the moderators, coloured in blue. As it was stated before, each comment is inspected by five crowdworkers. While most of the comments are unanimously classified as being not abusive (0 rejections), we observe that only a minority of workers rejected a large proportion of comments. In contrast, the remaining ones classified them as not abusive (e.g., 28,833 comments are only rejected by one crowdworker). In total, only 419 comments are unanimously classified as abusive language by 5 participants. Interestingly, if we compare those numbers to the moderators' rejection decision (indicated as blue bars), we see that the higher the rejection agreement of the crowdworkers, the higher the consensus with the moderators. Given these insights, we deduce that we can reduce the number of false positives described before by introducing a rejection agreement threshold (e.g., $\geq 2$) in the final dataset. As we already indicated previously: abusive language is considered a highly subjective term with no upon agreed definition. Thus, it is not surprising and in accordance with previous findings, that this ambiguity is also present when we inspect the inter-annotator agreement [57]. Calculating Krippendorff's alpha [31] for all of the 85,000 comments we find low agreement ($\alpha = 0.19$). This generally low agreement in the domain of abusive language detection is reported in multiple studies (e.g., [48] or [63]) and is often described as one of the more important challenges in this field [58]. To overcome this drawback for our new dataset, it is possible to adjust (increase) the mentioned agreement threshold. We have to emphasise that there exists a trade-off between available training data for underlying machine learning models and inter-annotator agreement, as depicted in Figure 2 (c). While there is a perfect annotator agreement for an agreement threshold of 5, this leaves only 419 comments in the rejection category.

Table 4: `RP-Mod` and `RP-Crowd` Structure

| Column | Description | Datatype | Ranges | RP-Mod | RP-Crowd |
|--------|-------------|----------|--------|--------|----------|
| ID | Unique identifier | `int` | - | * | * |
| Text | Text of the comment | `text` | - | * | * |
| Reject Newspaper | comment is rejected by moderators | `bool` | {0,1} | * | |
| Reject Crowd | comment is rejected by crowdworkers (majority decision) | `bool` | {0,1} | | * |
| Rejection Count Crowd | total number of rejections by crowd | `int` | [0,5] | | * |
| Sexism Count Crowd | # rejects by crowdworkers based on sexism | `int` | [0,5] | | * |
| Racism Count Crowd | # rejects by crowdworkers based on racism | `int` | [0,5] | | * |
| Threat Count Crowd | # rejects by crowdworkers based on threats | `int` | [0,5] | | * |
| Insult Count Crowd | # rejects by crowdworkers based on insult | `int` | [0,5] | | * |
| Profanity Count Crowd | # rejects by crowdworkers based on profanity | `int` | [0,5] | | * |
| Meta Count Crowd | # rejects by crowdworkers based on meta | `int` | [0,5] | | * |
| Advertisement Count Crowd | # rejects by crowdworkers based on advertisement | `int` | [0,5] | | * |

# 5 Experiments

In this section, we provide insights into the usefulness of the dataset to detect abusive comments in a classification setting. We compare well-known baseline approaches with different text representations as well as recent transformer-based language models. Moreover, we cross-evaluate our trained models on an existing dataset (`DerStandard`) published and analysed in [52, 51] to evaluate whether the models trained on our dataset generalise well and to illustrate the additional challenges imposed by our dataset. As the annotation schemes between the datasets are highly diverse, we consequently restrict our experiments to a binary classification setting (abusive or non-abusive).

## 5.1 Experimental Setup

One of our annotated dataset's major advantages is that five different crowdworkers labelled each comment. Naturally, we can use the decision count as a threshold for determining whether a comment should be labelled as abusive or not. Intuitively, it could be argued that only comments should be labelled as abusive if all annotators agree on this label. However, as indicated in Figure 2, the number of observations drastically decreases with more agreements, preventing the training on a sufficient amount of data. On the contrary, labelling a comment as abusive only based on one annotator's decision would lead to many false positives. In our experiments we train models on datasets with two different agreement thresholds: 2 and 3 (`RP-Crowd-2` and `RP-Crowd-3`). Moreover, we create a model that is solely based on the moderators' decision in `RP-Mod`. As the number of abusive comments is significantly smaller than unproblematic ones, we use undersampling to achieve a balanced label distribution for each of the mentioned datasets. Moreover, we rebalanced each of the mentioned datasets independently.

## 5.2 Baseline Classifiers

In context of the supervised binary baseline setting, we utilise two different text representations: frequency-weighted `tf-idf` vectors [50] and pre-trained `fasttext` embeddings [6]. For `tf-idf`, we restrict the maximum number of features to 3,200 and ignore rare terms that occur less than five times in all documents. Moreover, we use uni- and bigrams to take small context windows into account. The `fasttext` embeddings are pre-trained on a large corpus from Wikipedia and Common Crawl consisting of 300 dimensions. All texts are preprocessed before they are transformed into their respective representation. The preprocessing procedure includes stopword removal, lemmatisation, and lower-case transformation. As baseline algorithms, we utilise supervised methods proven to perform well on textual classification tasks (including abusive language detection). We use `Multinomial Naive Bayes`, `Logistic Regression`, `Gradient Boosted Trees` and `AutoML` pipelines[10]. Since embeddings are not count-based, we rely on the `Gaussian Naive Bayes` in that case. We use Bayesian hyperparameter optimisation[11] with 10-fold cross-validation[12] for all of our baseline models. We repeat each baseline experiment five times. For the construction of our AutoML pipelines, we set an overall time limit of 5 hours per run.

## 5.3 Transformer

In addition to our baseline classifiers, we fine-tune three BERT (Bidirectional Encoder Representations from Transformers) models [18] on each of the data-sets: `RP-Crowd-2`, `RP-Crowd-3` and `RP-Mod`. We fine-tuned three different pre-trained BERT models that were explicitly trained on the German language [10], each for for 100 epochs: $BERT_{Base}$ [13], $BERT_{Hate}$ [14] and $GBERT_{Base}$ [15]. The used tokenizers are also provided together with the pre-trained BERT models. For fine-tuning, we followed two approaches. First, we added an additional classification head on top of our pooled BERT output. The head consists of two dense layers with GELU (Gaussian error linear units) [26] activation in-between and a Softmax activation after the last layer. In addition, we used the cross-entropy loss. In

---

[10] https://automl.github.io/auto-sklearn/master/

[11] https://scikit-optimize.github.io

[12] The code for the experiments is available at https://github.com/Dennis1989/RP-Mod-RP-Crowd

[13] https://www.deepset.ai/german-bert

[14] https://huggingface.co/deepset/bert-base-german-cased-hatespeech-GermEval18Coarse

[15] https://huggingface.co/deepset/gbert-base

the following, we will refer to this method as `single-task` BERT. In a subsequent setup, we added a second training task in parallel to avoid overfitting and model instability [11] and also incorporate domain-specifics into the pre-trained language model [12]. Therefore, we again fine-tuned each BERT model to predict `abusive` or `non-abusive` as a simple, binary classification task. At the same time, we train the model on parts of the original language modelling objective, in which the model has to predict the original value of randomly masked tokens. Hereby, we followed closely the approach by Devlin et al. [18] in which $15\%$ of the tokens are masked. However, not all tokens are replaced by `[MASK]`, but $10\%$ are replaced by a random token and $10\%$ are not replaced. The two loss-functions are combined into a single loss-function, $\mathcal{L}_T$, with $\alpha$ as a weighting-factor:

$$\mathcal{L}_T(x, y, y_{LM} \mid f_\Theta) = \alpha \cdot \mathcal{L}_{LM}(x, y_{LM} \mid f_\Theta) + (1 - \alpha) \cdot \mathcal{L}_{CE}(x, y \mid f_\Theta)$$

Here, $f_\Theta$ is the model and $\Theta$ are its weights, while $\mathcal{L}_{CE}$ is the cross-entropy between the predicted label and the true label for `abusive` or `non-abusive` and $\mathcal{L}_{LM}$ is the loss-function for the language modelling task. We trained different models on different settings of $\alpha \in \{0.1, 0.5, 0.9\}$. We refer to this variant as `multi-task` BERT.

As recently suggested by [38], we furthermore applied the following steps to avoid instability during fine-tuning: First, we gradually unfreeze the pre-trained transformer block of the BERT model one by one, every five epochs. Last, we progressively decrease the `learning rate` – starting with $0.0001$ and multiply it by $0.97$ every epoch. We find that this setup resulted in the best overall performance on the three datasets. We conducted all of our experiments on three NVIDIA Quadro RTX 6000 with 24GB of memory.

### 5.4 Experimental Results

Figure 3 reveals performances of the best models as precision-recall (PR) and receiver operator characteristic (ROC) curves for all three evaluation datasets. For the PR curves, we utilised non-linear interpolation, proposed by Davis & Goadrich [16]. To no surprise, and in accordance with recent findings, the transformer-based BERT architecture (especially the newest GBERT$_{Base}$ model) significantly outperforms our baseline methods on all datasets. In general, we observe that the performance of the best BERT model (GBERT$_{Base}$, `multi-task`, $\alpha = 0.9$) trained on `RP-Crowd-3` (Area Under the Receiver Operating Characteristic curve: AUROC = 0.914) is higher than those trained on `RP-Crowd-2` (AUROC = 0.889) and both of them achieve higher scores than the best models trained on `RP-Mod` (AUROC = 0.791). We have to stress that this does not necessarily imply that the moderators' decisions are more flawed than the crowdworkers' and that these scores cannot be directly compared with each other. It is rather an indicator that the moderators' annotations and the resulting different data distributions are harder to be modelled by a supervised algorithm. Respective reasons can be manifold, such as more complex personal intrinsic rejection rules based on previous knowledge (moderators often know from which accounts comments should be carefully monitored). However, a clear causal explanation cannot be made and leaves room for upcoming research. For our baseline approaches, word-embedding representations generally perform slightly better than their `tf-idf` counterparts (transparent vs. non-transparent curves). Here, an exception is the `Naive Bayes` approach. However, as mentioned before, `Gaussian Bayes` was used in the context of a word-embedding representation, resulting in poor performance.

During the training of the `single-task` BERT models, we experienced a model-deterioration to a random classifier (Val Acc: 50%), similar to what [38] describe in their work. This model failing was not present in any of the `multi-task` settings. Examples of model instability on single-task models can be found in the Appendix. It is evident that both losses (train and validation) seem to converge during model training, implying that 100 epochs are sufficient for the current setting. According to our results, a low $\alpha$ results in high accuracy variability during training and a tendency of overfitting, while this is mitigated with higher $\alpha$ settings. Further details on model convergence for different $\alpha$ settings can be also assessed in the Appendix.

In the domain of comment moderation, it is essential to emphasise the different importance of classification errors. While in many traditional supervised settings, both false negatives, as well as false positives, play an equally important role, models that are used in the context of comment moderation focus more on reducing the number of false negatives (comments that are wrongly classified as non-abusive although they do contain inappropriate content). The rationale behind this is that comment moderation is usually not completely automated. In this setting, machine learning models are used as a pre-filter for comments which should be analysed and checked by a moderator;

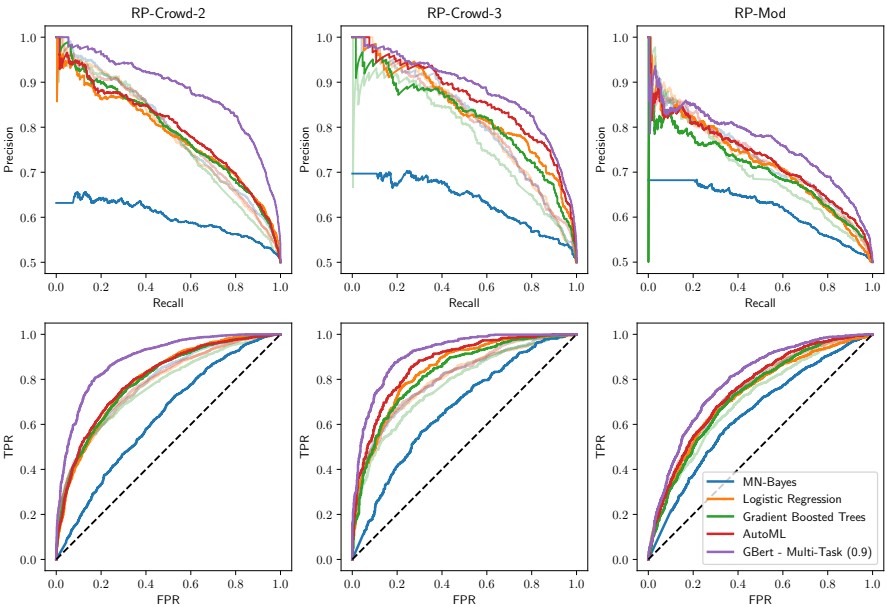

Figure 3: Precision-Recall (top) and ROC curves (bottom) for the best model instances on all datasets. Transparent curves → `tf-idf` representations; non-transparent curves → `fasttext` embeddings

all other comments are published directly. Thus, it is preferable to make more rigorous decisions than directly publishing comments that may contain abusive language. Therefore, in this particular instance, maximising recall is prioritised over maximising precision. In other use-cases, this might be different. Hence, we find it important to report PR and ROC curves. While extreme thresholds are not desirable, it is up to the concrete scenario to determine a suitable configuration.

In the context of our cross-evaluation, we only focus on BERT models, as they outperform all other baseline approaches. To perform the cross-evaluation, we train an additional $\text{GBERT}_{Base}(\alpha = 0.9)$ on the `DerStandard` dataset with the same setting as described before. Subsequently, we use all three $\text{GBERT}_{Base}(\alpha = 0.9)$ models trained on our dataset instances (`RP-Crowd-2`, `RP-Crowd-3` and `RP-Mod`) and individually evaluate them on the validation-split of `DerStandard` dataset. In the other direction, we evaluate the $\text{GBERT}_{Base}$ model trained on `DerStandard` on the test splits of `RP-Mod`, `RP-Crowd-2` and `RP-Crowd-3`. We also perform this cross-evaluation for the different RP models. Results can be inspected in Figure 4.

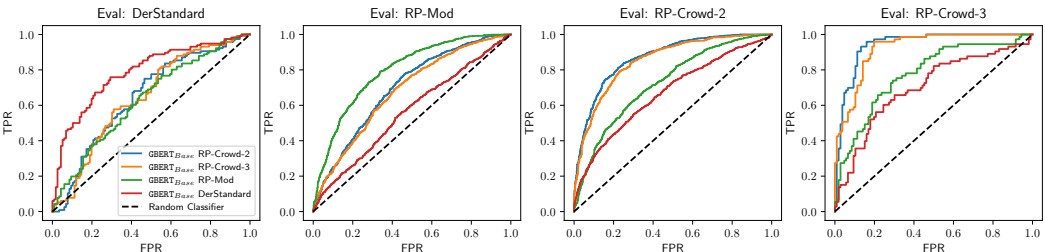

Figure 4: Cross-evaluation of our best models on different datasets. From left to right: cross-evaluation on `DerStandard`, `RP-Mod`, `RP-Crowd-2`, and on `RP-Crowd-3`

The two leftmost subplots in Figure 4 visualise the ROC performances of our models evaluated on `DerStandard` and `RP-Mod` (both datasets were annotated by professional moderators). To no surprise and in line with recent cross-evaluation studies, it seems that the models do not generalise well to different domains. While models trained on our RP data produce results clearly above random guesses on `DerStandard` (AUROC = 0.65 for $\text{GBERT}_{Base}$ on `RP-Crowd-2`), the evaluation of the `DerStandard` model on `RP-Mod` results in a ROC curve that is close to a random classifier. Another

interesting observation is that cross-evaluation on our crowd-annotated datasets (two rightmost subplots) in general produced higher performance results. This again is an indicator that the underlying decision distribution of the crowd can be easier modelled than the professional moderator's. Quite possibly, the crowd datasets consist of comment instances which contain *obvious*, easy to identify abusiveness. We leave this investigation up to future research endeavours. The underlying reason for the poor overall performance of models trained on `DerStandard` in general is that the number of ground-truth observations for this dataset is relatively small compared to our data (only 1,060 relevant observations). We again highlight that for both cross-evaluation directions in general, the models do not perform well, similar to previous studies [29, 64]. Moving towards more generalisable models is an important research branch in the area of abusive language detection (cf., [61, 24]). Our new dataset enables researchers to explore the underlying causes of poor generalisability (e.g., topical shifts) now for the German language, which is, even more, a testament to the importance of our new dataset to the community (cf., [61] who emphasize the necessity for cross-domain evaluation and cf., [57] for the concerning lack of comparison data for many languages).

## 6 Discussion

With this publication we try to address two of the issues raised by [57] and multiple other researchers [13, 36], that currently, the field of abusive language detection experiences data scarcity and "relies too heavily on data from Twitter" [57]. Towards remedying these issues, we contribute the largest, labelled German dataset of online newspaper comments – which would otherwise not be accessible – and make it publicly accessible for any non-commercial use with this work. Each comment of the dataset is annotated by moderators of the RP and five crowdworkers, allowing for the creation of different ground-truth variants. We present essential characteristics of our new dataset and elaborate on the demographic properties of our crowd-study participants. Our experiments show that our dataset is suitable for training models to identify abusive language in a binary classification setting. By utilising transformer-based language models, high classification standards can be achieved.

**Ethical Considerations**    The datasets consists of comments that were created by real humans and by purpose contain abusive content. Users who are sensitive to such language should handle the dataset cautiously. We ensured that users were made aware of the fact that their comments are used for scientific purposes by adjusting the Terms of Service prior to data collection. To avoid introducing biases [15], all comments of the RP have been considered without further topical filtering. Furthermore, using a broad crowd to annotate the data should minimise the inclusion of person-specific biases. The dataset can be used to build automated moderation systems, with implications to the freedom of expression. Hence, the dataset should only be used for decision support systems, or automated acceptance.

**Limitations & Outlook**    As already indicated, there is room for further experimentation on our new dataset. Up until now, only the binary classification setting was analysed. Since we provide a more granular annotation scheme, future endeavours should focus on a multi-class and -label classification setting. Moreover, the discrepancy between the moderator's and crowd annotations should be subject to a thorough investigation that may lead to creating an even more accurate ground truth and domain understanding. Combining the loss of the language modelling with the classification task proved to be promising and resulted in less overfitting and fewer model instabilities. However, this is not a benchmarking paper and only serves as a starting point for developing more sophisticated model architectures that may even incorporate additional auxiliary features such as the output of a sentiment analysis. In the long run, especially to increase generalisability, even more and much larger datasets of this type are needed, which ideally will be supplemented and adapted over time. Ultimately, this knowledge will foster the development of (semi-)automatic comment moderation platforms and facilitate a fair online discussion culture.

## Acknowledgments

The research leading to these results received funding from the federal state of North Rhine-Westphalia and the European Regional Development Fund (EFRE.NRW 2014-2020), Project: M●DERAT! (No. CM-2-2-036a). We would further like to thank Hannah Monderkamp, Jens Brunk, and Jörg Becker for their assistance in the early stages of the dataset creation.

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
