# OpenReview forum: "$\texttt{RP-Mod}\ \&\ \texttt{RP-Crowd:}$ Moderator- and Crowd-Annotated German News Comment Datasets"
_NeurIPS.cc/2021/Track/Datasets_and_Benchmarks/Round2 — NeurIPS 2021 Datasets and Benchmarks Track (Round 2)_

### Official Review · Reviewer_y8tm · 2021-09-14
**Problems in the evaluation part and possible ethical problems**

**Rating:** 6
**Confidence:** 3

**Strengths:**

New data set, not English.

**Weaknesses:**

I am not sure if the authors paid enough attention to the reliability of crowd workers.
They certainly followed standard protocols: attention checks and gold standard questions.
Nevertheless it is IMHO a known fact that crowdworkers will have a strong tendency to choose the route of least resistance, i.e. to provide easy answers. This is a big issue in current NLP. For example when generating question-answering data sets, you are likely to get trivial questions such as "Who did x" that take the least effort to formulate.
This needs to be taken into account here. For example, a post can be abusive in multiple ways - the current assignment only allowed for single tagging. It also mixes two questions: (1) in which way the post is abusive, and (2) whether it passes the threshold that it should be withheld by the moderators. A post could be abusive (e.g., Angela Merkel is <insult>) but it may not be necessary to remove it.
Hence (see also the comments below) I am not convinced that the data quality of this data set is good enough to be "suitable for training models to identify abusive language in a binary classification setting" - the models trained on this data perform poorly on the DerStandard data set, i.e., it does not appear to transfer to other data sets. It may be a suitable data set for experimentation with such models, but there may be a high risk of overfitting to particularities of this data set.

One may well argue that this is up to the user of the data set to decide and experimentally verify, though. And in general I remain in favor of accepting this data set paper, if it weren't for the problems discussed in the ethics section.

Page 7, line 248: you use PR and ROC curves, but then give only one AUC value. Below which of the two curves?
Was there any train-test split used in this experiment?

Page 7, line 253. Here you effectively compare AUC scores on two different data sets, RP-Crowd-2 and RP-Crowd-3. But if you change the threshold when comments are considered abusive or not, then it means the data distributions changes. The same holds for RP-Mod - comparing apples and oranges.
More precisely, it is very likely that RP-Crowd-2, RP-Crowd-3 and RP-Mod agree on easy instances, and differ on difficult cases; and changing the threshold can make the data set easier or harder. Because of this issue, it may be insightful to cross-evaluate: how well does the RP-Crowd-2 models perform with RP-Crowd-3 and RP-Mod evaluation.


**Additional Feedback:**

The authors should reconsider the placement of Figure 3 in between of two paragraphs, neither of which relates to Figure 3.

For the format of this submission, I would have expected more detailed documentation of the data labeling, and less focus on the preliminary experiments (which also are not a strength of the paper...)

**Clarity:**

The authors are unclear on some issues in their evaluation procedure, such as which AUC they used, and how this measure relates to the problem to solve.

I think the annotation process by the crowdworkers should have been documented much better in the actual paper, not only the appendix.
- were crowd-workers asked a binary "problematic or not" question, or did they have to choose the label? i.e., the "full" labeling scheme in Table 2, was it used for the crowd workers, or not?
- if annotators used different labels from this table, how was a final labeling reached as there might not be a majority label?
- were any crowd worked labels discarded, because they were identified as unreliable? For example the "outlier" crowd workers that flagged up to 85% of comments as abusive likely should NOT be used. Is this information available for users, or does the published data set only include the labels obtained via pre-defined thresholds?
- it was not clear to me when reading the paper that RP-2 and RP-3 were besides setting the threshold differently, also re-balanced independently.

**Correctness:**

Page 8, the P-R-Curves clearly are incorrectly generated. First of all, the curves must not be assumed to begin at (0,1) - in fact, the value at 0 is undefined - as this will lead to overestimation of the performance. The best guess for a starting point (0,y) is the y of the first actual sample. Secondly, the curves must not be linearly interpolated, instead an exponential interpolation is necessary. In particularly the blue lines are affected by this.

For further details on incorrect interpolation on PR-curves, see: Davis, Jesse, and Mark Goadrich. "The relationship between Precision-Recall and ROC curves." Proceedings of the 23rd international conference on Machine learning. 2006.

Also note that AUC is not adjusted for chance. For ROC, a random rankings scores 0.5, hence the difference between 0.78 and 0.67 is after adjustment to chance the difference between 0.56 and 0.34, a quite substantial drop.

Do not just follow a simple number aggregate such as the AUC. It is too coarse, and it does not relate to the problem that you are trying to solve. The task that you are solving is to filter out abusive comments. This means that you will likely need to allow a quite small false positive rate, while you still need a fairly high true positive rate.
The AUC aggregates average over all possible recall respectively FPR values. But high FPR values or low recall values are useless systems.
Consider that we have 10% abusive comments, and we want to falsely flag at most 10% of comments incorrectly. This means that for the ROC curve, only the leftmost part of the curve (up to 10%) is of interest, any performance further to the right is moot. Figure 5a clearly indicates that all the RP-trained classifiers perform little different from random in this range. It is interesting to see that 5b differs very clearly, the DerStandard trained models seem to work okayish for the RP data. Why?
In line 285 you claim that the models trained on RP-Crowd-2 performed best on the RP-Crowd-3 data. But this exactly due to the evaluation problem! If you consider the PR-Curve on the left side of 5b, you can see that the RP-Crowd-3 model performed significantly better up to a recall of 0.25. But these early detections are exactly where such a system would be potentially useful, not the long tail on the right!

In line 287, you argue that "the underlying reason for poor cross-evaluation performance of models trained on DerStandard is ..." - I beg to disagree: the DerStandard trained model transferred to new data much better than the models trained on RP-Crowd, which performed really poor on the DerStandard data. This conclusion is wrong, in my opinion. And while I agree that there is a high importance for additional training data, my concern is that the provided crowd-sourced-data may not have sufficient quality to help much: Training on the new data set does not transfer to DerStandard data, hence the models may have learned particular data-set specific signals (e.g., if it contains "MAGA", then it is likely bad).

DUPLICATES have been assigned to different folds. This causes overestimation of the quality. Sometimes they also have different labels. Probably not a huge amount. Some 500 to 1000 (near-) duplicates I estimate, so 1%-2%. Nevertheless, these pairs should always be assigned the same fold.

**Documentation:**

Detailed documentation, but some parts should IMHO have been included in the main description.

To fully be able to judge if commenters have agreed to this usage of their comments, the legal code they agreed to would need to be included, not only that for the annotators.

**Ethics:**

The instructions to the participants (c.f., page 15) include the following statement.
"Please note: Your submitted data will only be used for the purposes of the research project at the University of Münster."
This does IMHO not include making them publicly available.
You promised they are *only* used by your research project.

On page 26 it is stated that commenters "knowing that their texts might be subject to academic use and republishing." - while the current intended CC-BY-NC-SA license prohibits commerical use, it is not strictly limited to academic/research use.

Also it is not obvious that the comment authors have the IP rights on what they posted. In particular flagged comments could well include copyrighted material (which is problematic). E.g., what if one would copy parts of Harry Potter as comments? Such contents could be contained in this data set. People that post abusive content are likely not following IP rights well either.

"Does the dataset relate to people?" - while you do not include the identities of the commenters, the comments might relate to people. After all, some of these comments are insulting and otherwise abusive. Some of the comments themselves might be legally problematic to publish! What if there were a comment "<First Author> is a <insult> and lives at <home address>"? I do not think comments can easily be republished.

In A.4.1, there are eight people named that were "involved" "throughout the creation of the dataset". Of these, three were not made authors of the data set article, why are they not even credited? And the name that is not "involved", what is his contribution to the data set article?

**Relation To Prior Work:**

Good.

**Summary And Contributions:**

The authors offer a new data set for comment moderation, in German. The data is partially crowd-sourced with the usual drawbacks (annotator reliability). An initial experimental study is conducted with baseline classifiers.
There may be an ethical problem, as the crowdworkers were apparently instructed that their answers are *only* used by the university research project, and now they are made publicly available. Furthermore, it is unclear if the actual comments are all legal, as abusive and problematic comments can, for example, include copyright violations and personally identifiable information.
There is also an evaluation problem (too much about optimizing some evaluation score, not about solving an actual problem), but that part of the paper is not central.

---

> ### Author Response · Authors · 2021-09-28
> **Author Response I**
>
> Thank you very much for your comprehensive review! We greatly appreciate your work and carefully reviewed all of your insightful comments. In the following, we are going to comment on all of the questions and issues you have raised. We hope that our elaborations help in clarifying your concerns.
>
> > Weaknesses:
> I am not sure if the authors paid enough attention to the reliability of crowd workers. They certainly followed standard protocols: attention checks and gold standard questions. Nevertheless it is IMHO a known fact that crowdworkers will have a strong tendency to choose the route of least resistance, i.e. to provide easy answers. This is a big issue in current NLP. For example when generating question-answering data sets, you are likely to get trivial questions such as "Who did x" that take the least effort to formulate. This needs to be taken into account here. For example, a post can be abusive in multiple ways - the current assignment only allowed for single tagging. It also mixes two questions: (1) in which way the post is abusive, and (2) whether it passes the threshold that it should be withheld by the moderators. A post could be abusive (e.g., Angela Merkel is <insult>) but it may not be necessary to remove it.
>
> By conducting a pre-screening and through the interleaving of attention-checks and gold-standard comments we indeed tried to follow the standard protocols as recommended in the related literature. While we agree that crowdworkers (as most humans) will try to optimise their earnings, an analysis of the assigned labels does not indicate that people tried to sort everything into the first reasonable category.
>     Furthermore, there seems to be a misunderstanding about the annotation procedure. Our crowdworkers had to tag each comment (that they wanted to reject) with at least one label; however, they could assign multiple labels if they deemed this necessary. Hence, our dataset has a multi-label annotation (except the Rejection vs. Acceptance decision which is inherently binary). After carefully re-reading the corresponding Sections 3.3 and 4 in our paper, we added additional explanations to point out that the annotation allowed for the assignment of multiple labels.
>
> We do agree, that the line between abusive language and the moderation decision is indeed a blurry one. With our dataset, we try to contribute a corpus of decisions made from multiple points of view. Even professional moderators do not necessarily always agree, either on the amount of abusiveness within a comment or on the final moderation decision. Thus, we only provide the opinions of the crowdworkers who participated in the study. The final moderation decision (removal or not) is up to the threshold implementation of the researcher.

---

> ### Author Response · Authors · 2021-09-28
> **Author Response II**
>
> >Hence (see also the comments below) I am not convinced that the data quality of this data set is good enough to be "suitable for training models to identify abusive language in a binary classification setting" - the models trained on this data perform poorly on the DerStandard data set, i.e., it does not appear to transfer to other data sets. It may be a suitable data set for experimentation with such models, but there may be a high risk of overfitting to particularities of this data set.
>
> Considering the complexity of the abusive language detection domain we understand your concerns. Despite almost two decades of work, no optimal solution has been identified, even for the more intensively researched English language. However, we are confident that the provision of our dataset is an important step towards improving abusive language detection models, as one of the most pressing issues is the lack of labelled data.
> Furthermore, we would like to contextualise the poorer performance on the DerStandard dataset:
> * First, the DerStandard dataset is older (2017 vs. 2018-2020) which has an impact on both the language used as well as topics discussed (topical shifts change the data distribution).
> * Second, it is sourced from a different country which has a considerable impact on the discussed topics as well, as, e.g., political affairs differ.
> * Third, the generalisation performance in the abusive language detection domain has been identified to be problematic already in 2018 by [1] and more recently by [2].
> * Fourth, we would like to have evaluated our solution with additional German newspaper comment datasets to have more generalisable results; however, the DerStandard dataset is the only one currently known to us.
>
>         [1] Karan, M., and Šnajder, J. 2018. “Cross-Domain Detection of Abusive Language Online,” in Proceedings of the Second Workshop on Abusive Language Online, ALW2, D. Fišer, R. Huang, V. Prabhakaran, R. Voigt, Z. Waseem, and J. Wernimont (eds.), Brussels, Belgium: Association for Computational Linguistics, pp. 132–137.
>         [2] Yin, W., and Zubiaga, A. 2021. “Towards Generalisable Hate Speech Detection: A Review on Obstacles and Solutions,” PeerJ Computer Science (7), pp. 1–38. (https://doi.org/10.7717/PEERJ-CS.598).
>
>
> Overfitting on particularities of datasets is, in general, something that has to be avoided during the model training and is not necessarily only a problem of the dataset itself. This is something we observed within both of our cross-evaluation experiments. Future research might investigate those tendencies and explore the performance of mixed datasets. However, in the context of this paper, we are more focussed on presenting the new dataset.
>
> > One may well argue that this is up to the user of the data set to decide and experimentally verify, though. And in general I remain in favor of accepting this data set paper, if it weren't for the problems discussed in the ethics section.
>
> We acknowledge that in our initial version of the paper, there have been some imprecision and uncertainties regarding the underlying ethics of the data collection. However, we used the opportunity of this review to improve the corresponding parts of our paper. The details are outlined in the following.
>
> > Page 7, line 248: you use PR and ROC curves, but then give only one AUC value. Below which of the two curves? Was there any train-test split used in this experiment?
>
> The AUC values that are presented in the paper were calculated from the ROC curves. We understand the confusion and addressed this issue in the paper by now unambiguously referring to AUROC. We calculate the AUROC values by using a test set that was not used during training.

---

> ### Author Response · Authors · 2021-09-28
> **Author Response III**
>
>
> > Page 7, line 253. Here you effectively compare AUC scores on two different data sets, RP-Crowd-2 and RP-Crowd-3. But if you change the threshold when comments are considered abusive or not, then it means the data distributions changes. The same holds for RP-Mod - comparing apples and oranges. More precisely, it is very likely that RP-Crowd-2, RP-Crowd-3 and RP-Mod agree on easy instances, and differ on difficult cases; and changing the threshold can make the data set easier or harder. Because of this issue, it may be insightful to cross-evaluate: how well does the RP-Crowd-2 models perform with RP-Crowd-3 and RP-Mod evaluation.
>
> This is one important aspect that you raised and you are abolutely correct with your statement that the AUROC scores resulting from RP-Crowd-2, RP-Crowd-3, and RP-Mod are not directly comparable as the models are trained on completely different data distributions and you are also right that these distributions probably represent different task-difficulties (more restrictive -> easier to detect). This was misleading in the original submission and was clarified in the current version. So why did we report the metric at all? In general we would like to give the reader an idea of what to expect from our dataset, when trained on state-of-the-art language models. We agree with your later comments that this single metric does not tell the whole story, as it reflects a mean of multiple thresholds of which some are not really useful (see the discussion below). However, as most current papers report the AUROC, it provides an intuitive understanding to the ML-researcher of what performance to expect. Of course it is not good practice to just rely on one single metric value. Therefore we reported PR and ROC curves for all scenarios.
>
> We will cross-evaluate our RP-Crowd-2 models on RP-Crowd-3 and RP-Mod and vice-versa. However, as you correctly pointed out, there are duplicates assigned to different folds, we first have to solve the duplicate issue before cross-evaluating the models without these overestimation issues. Moreover, it is currently possible that during cross-evaluation test instances of models trained on RP-Crowd-2 are actual training instances for models trained on RP-Crowd-3 (and vice versa) again resulting in overestimation of the cross-validation performance. We will address these small tendencies of overestimation by removing those observations during cross-evaluation and incorporate the results in the final camera-ready version.
>
> > Correctness:
> Page 8, the P-R-Curves clearly are incorrectly generated. First of all, the curves must not be assumed to begin at (0,1) - in fact, the value at 0 is undefined - as this will lead to overestimation of the performance. The best guess for a starting point (0,y) is the y of the first actual sample. Secondly, the curves must not be linearly interpolated, instead an exponential interpolation is necessary. In particularly the blue lines are affected by this.
> For further details on incorrect interpolation on PR-curves, see: Davis, Jesse, and Mark Goadrich. "The relationship between Precision-Recall and ROC curves." Proceedings of the 23rd international conference on Machine learning. 2006.
>
> We would like to sincerely thank you for this remark and also for directly referring us to the relevant literature. Frankly, we were not aware of this interpolation issue and are not only happy to correct the PR-curves in this publication, but also know better for future research endeavours. To calculate the new PR curves we now use an R package "PRROC" (https://cran.r-project.org/web/packages/PRROC/index.html). Also you were right that the blue lines in particular are affected, since they also are no longer assumed to begin at (0,1). While there are no other significant changes in the curves, especially the differences between the AutoML and other baseline classifiers are now more pronounced. Again, thank you very much for this valuable advice.

---

> ### Author Response · Authors · 2021-09-28
> **Author Response IV**
>
> > Also note that AUC is not adjusted for chance. For ROC, a random rankings scores 0.5, hence the difference between 0.78 and 0.67 is after adjustment to chance the difference between 0.56 and 0.34, a quite substantial drop.
> Do not just follow a simple number aggregate such as the AUC. It is too coarse, and it does not relate to the problem that you are trying to solve. The task that you are solving is to filter out abusive comments. This means that you will likely need to allow a quite small false positive rate, while you still need a fairly high true positive rate. The AUC aggregates average over all possible recall respectively FPR values. But high FPR values or low recall values are useless systems. Consider that we have 10% abusive comments, and we want to falsely flag at most 10% of comments incorrectly. This means that for the ROC curve, only the leftmost part of the curve (up to 10%) is of interest, any performance further to the right is moot. Figure 5a clearly indicates that all the RP-trained classifiers perform little different from random in this range. It is interesting to see that 5b differs very clearly, the DerStandard trained models seem to work okayish for the RP data. Why? In line 285 you claim that the models trained on RP-Crowd-2 performed best on the RP-Crowd-3 data. But this exactly due to the evaluation problem! If you consider the PR-Curve on the left side of 5b, you can see that the RP-Crowd-3 model performed significantly better up to a recall of 0.25. But these early detections are exactly where such a system would be potentially useful, not the long tail on the right!
>
> We appreciate your remark and your really sophisticated and detailed analysis of our evaluation results. To address your concerns, we would like to elaborate on how the trained models will be utilised in practice. When those models are deployed on the newspaper's side, the main goal is to provide a semi-automatic support during their moderation process. In this particular use-case false negatives are considered worse than false positives, as those comments which are falsely classified as abusive (FP) are manually inspected by the moderators for a final decision (leading to more workload). In contrary, false negatives are comments that will automatically be published within the comment section although they contain abusive language, resulting in negative external impression, which is considered much worse than more workload. Hence, it was communicated by our collaboration partner multiple times that FNs are errors that have to be avoided, while FPs can be accepted to a certain degree. So we are especially interested in models with high Recall. While we agree that extremely high FPR values are useless, we think that far more than 10% of the left side of the ROC curve are relevant thresholds and that not only the super early detections are the only scenario were these systems are useful. Hence, we leave the question of which threshold should be chosen to upcoming research (we think that identifying kinks in the PR curves might be beneficial). However, we explicitly addressed this issue in the paper (see Section 5.4) and describe that depending on the use-case at hand it is necessary to put emphasis on different errors (either FPs or FNs).
>
> > In line 287, you argue that "the underlying reason for poor cross-evaluation performance of models trained on DerStandard is ..." - I beg to disagree: the DerStandard trained model transferred to new data much better than the models trained on RP-Crowd, which performed really poor on the DerStandard data. This conclusion is wrong, in my opinion. And while I agree that there is a high importance for additional training data, my concern is that the provided crowd-sourced-data may not have sufficient quality to help much: Training on the new data set does not transfer to DerStandard data, hence the models may have learned particular data-set specific signals (e.g., if it contains "MAGA", then it is likely bad).
>
> For the leftmost (below 10%) part of the ROC curve your statement is completely correct. However, as we already argued in our preceding answer, we do not agree that 10% is necessarily a good threshold. A higher FPR means more workload for the moderators, but less FN. Therefore, depending on the acceptable workload of the moderators, one can argue to allow 20% or even 30% FPR. In this scenario, RP-Crowd-2 and RP-Crowd-3 perform well, better than DerStandard.
> Nonetheless, our phrasing in the original paper is also not completely adequate. We, therefore, decided to highlight that both cross-evaluation experiments do not perform really well, which is something that is frequently observed in the domain of abusive language detection and might be due to different topic distributions (not only because of the time difference but also because the datasets originate from different countries).

---

> ### Author Response · Authors · 2021-09-28
> **Author Response V**
>
>
> >DUPLICATES have been assigned to different folds. This causes overestimation of the quality. Sometimes they also have different labels. Probably not a huge amount. Some 500 to 1000 (near-) duplicates I estimate, so 1%-2%. Nevertheless, these pairs should always be assigned the same fold.
>
> Our dataset indeed contains some near- and real-duplicates. To be specific, in total we found 677 near-duplicates (Levenshtein distance with 0.15 treshold) and 815 real-duplicates which is 1.7% in total (again your prediction was correct). We also fully agree that the assignment of duplicates to different folds causes overestimation. We therefore have two options: (1) we can either assign the duplicates to the same folds or (2) remove them from the dataset and retrain the models. We will in the final camera-ready version report our results. However, we agree that it will not produce significantly different results.
>
> > Clarity:
> The authors are unclear on some issues in their evaluation procedure, such as which AUC they used, and how this measure relates to the problem to solve.
>
> As described before, we clarified these issues in our updated version and now correctly refer to area under the roc curve (AUROC).
>
> > I think the annotation process by the crowdworkers should have been documented much better in the actual paper, not only the appendix.
>
> We do agree with your statement. We initially decided to include a significant portion of the annotation process within the Appendix, as we were uncertain if a corresponding paragraph would add enough information for the reader. However, we do see the importance of said process for the main part of the paper. Therefore, utilising the additional page, we now address this issue more thoroughly by extending the description of the dataset creation process on page 4 f.
>
> > were crowd-workers asked a binary "problematic or not" question, or did they have to choose the label? i.e., the "full" labeling scheme in Table 2, was it used for the crowd workers, or not?
>
> For the creation of our annotations, we applied a hierarchical labeling approach (cf., Section 3.2 of our paper and Appendices A.1.1 and A.1.2) - so we did in fact ask our crowdworkers to do both:
>     First, for each comment the crowdworker has to evaluate, whether it should be published or not (cf., Figure 6 [Appendix A.1.2]).
>     If the crowdworker decided to reject a comment, at least one (potentially many) of the labels of our labeling schema (explained in Table 2) had to be assigned (cf., Figure 5 [Appendix A.1.2]).
>     Following your general request for more in-paper documentation, we added a corresponding paragraph to Section 3.3, also referencing Appendix A.1.1 and Figures 5-6 as sources of further information.
>
> > if annotators used different labels from this table, how was a final labeling reached as there might not be a majority label?
>
> We agree, that there is not always a majority vote for each comment in the dataset. However, the dataset has been conceptualised, presented and is supposed to be published with a binary and a multi-label labeling. For the binary case, the problem described breaks down to identifying an optimal threshold for rejecting respectively accepting a comment (as covered in experiments). The same applies to the multi-label case, where a threshold has to be chosen to decide which labels should be included, yet it is unproblematic if multiple labels have the same number of votes.
> The dataset can be converted into a multi-class dataset. However, this would require a strategy to identify the class of a comment, especially in cases without a majority vote. This is certainly interesting for future work with the dataset.

---

> ### Author Response · Authors · 2021-09-28
> **Author Response VI**
>
> We did not discard any labels assigned by the crowdworkers. As stated, abusiveness is a very subjective term, which can be interpreted in multiple ways. Within our dataset, we try to capture broad views on this topic. This means, that we will have people who tend to accept or reject comments more frequently than the "norm". However, the dataset does contain two outliers who flagged comments as abusive in an unusual amount of cases. During a closer inspection, in both cases, the assigned label was meta/organisational. Yet, it should be stated that the vast majority of these comments can be considered harsh enough to warrant rejection by a very sensitive person.
> This label does not directly refer to abusiveness per se, but to organisation specific criteria. As stated in the paper, we did not include the organisation-specific labels within our training and test data. In summary, the comments annotated by both outliers w.r.t. the theory-deduced labels are sound.
>
> > it was not clear to me when reading the paper that RP-2 and RP-3 were besides setting the threshold differently, also re-balanced independently.
>
> The re-balancing is now explicitly mentioned in the paper.
>
> > Documentation:
> Detailed documentation, but some parts should IMHO have been included in the main description.
>
> Similarly to your comment about the annotation process, we agree with your point. Thus, we have included Table 4 from the Appendix within the dataset description. Further, we elaborated more on the dataset characteristics within Chapter 4.

---

> ### Author Response · Authors · 2021-09-28
> **Author Response VII**
>
>
> > To fully be able to judge if commenters have agreed to this usage of their comments, the legal code they agreed to would need to be included, not only that for the annotators.
>
> Thank you for bringing up this issue. We agree that our documentation was probably a bit short on this. Indeed, the commenters have agreed to all possible kinds of use of their data as - by publishing their comments on the website of Rheinische Post - they had to agree to their terms and services which grants Rheinische Post a comprehensive, spatially, and temporally unlimited right of using and re-publishing their comments.
>
> The terms and services (translated into English) read as follows. Please note that we have uploaded the full terms and services including the English translation as well as the German original on [OpenReview.net](https://openreview.net/attachment?id=NfTU-wN8Uo&name=supplementary_material):
>
> "*If the user posts a copyrighted or otherwise legally protected contribution/text/pictures/graphics on RP ONLINE or on the special services and portals operated by RP ONLINE, e.g. in forums, guest books, comments, blogs, TONIGHT, etc., the user grants RP ONLINE a comprehensive, spatially and temporally unlimited right of use to the content for the corresponding dissemination on RP ONLINE and in the print edition of 'RP ONLINE'. If the user grants RP ONLINE a comprehensive, spatially and temporally unlimited right to use the content for further dissemination on RP ONLINE and in the print edition of the 'Rheinische Post', including the right to edit, redesign, and translate the content, to use it electronically/digitally, in any form and on any medium, and to publish it in book form.*"
>
> While this clearly allows using the comments for research purposes as well as publishing these comments in a dataset, as we do now, it should be noted that the users indeed did not agree to participate in this particular study. This could be questioned from an ethical, though not from a legal standpoint. We argue that the users have intentionally published their comments on Rheinische Post's website to be publicly visible to all visitors. Hence, the intention of the commentators was to have their comment(s) appear online. Consequently, the commentators were well aware that whatever they write would be visible to anyone on the internet. Please keep in mind that Rheinische Post is a publicly accessible website, i.e. not a closed community! Furthermore, if we mentioned our research project explicitly, we would receive a bias in our dataset, which cannot be statistically corrected, as the baseline remains unknown.
>
> Regarding abusive comments which have been "deleted" (i.e. hidden) by Rheinische Post, the situation remains the same. For this content, commentators still had the intention to publish it online. From our point of view there is no ethical issue with publication. To ensure that these comments cannot be traced back to individuals we are not including any user identifiers (eg. ID) in our dataset. Therefore, even by cross-comparing comments still visible online with abusive comments from our dataset, one cannot identify users.
>
> Lastly, while the terms and services mentioned above grant Rheinische Post the permission to publish and re-distribute the comments, Rheinische Post has explicitly granted us permission to publish the dataset here. This document is now included at [OpenReview.net](https://openreview.net/attachment?id=NfTU-wN8Uo&name=supplementary_material) as well.
>
> Consequently, we are only publishing comments of users who explicitly wanted their comments to appear online and we have - via Rheinische Post - all required permissions to publish the dataset.

---

> ### Author Response · Authors · 2021-09-28
> **Author Response VIII**
>
> > Ethics:
> The instructions to the participants (c.f., page 15) include the following statement. "Please note: Your submitted data will only be used for the purposes of the research project at the University of Münster." This does IMHO not include making them publicly available. You promised they are only used by your research project.
>
> This now refers not to the commentators, but to the crowdworkers who moderated our comments dataset. At this point, we would like to stress that this study was not a web survey with random participants, but a payed service, where the participats rated the comments as part of their payed job.
>
> As mentioned in our paper, the crowdworking was done by a subcontractor, a platform named CrowdGuru. CrowdGuru is a brand of IT2media GmbH & Co. KG, located in Germany. All crowdworkers working for CrowdGuru have signed a contract with IT2media to be part of their workforce. The concept is very similar to Amazon Mechanical Turk, except that IT2media as a Germany-based organisation was able to guarantee paying minimum wage to their crowdworkers. For both ethical and legal reasons, this was a requirement for us.
>
> Part of the contract between IT2media and their workers ("Gurus") is the following agreement, which grants CrowdGuru a comprehensive, exclusive, transferable, temporally, substantively, and spatially unlimited right of using the data provided by the workers. The statement (translated into English) reads as follows (again, the original can be found on OpenReview):
>
> "*IT2media is exclusively entitled to copyright-protected work results from the Guru's activities. The Guru undertakes to surrender and make available the results of his activities at any time upon IT2media's request, and to irrevocably grant IT2media the comprehensive, exclusive, transferable rights to the results of his work, which are unrestricted in terms of time, content and territory.*"
>
> Therefore, we have all required legal permissions on redistributing and publishing this data.
>
> Beyond this we did not publish any raw data acquired through the crowdsourcing study, but only our aggregated results derived from it. The published dataset will only contain the number of times a certain comment was voted for a specific label; however, we will (never) provide any information about who voted which comments. This information will remain confidential!
>
> The statement you are referring to on p. 15 was a sentence on the first page of the system, where crowdworkers had to rate the comments. This was only visible to the crowdworkers **after** they had chosen to participate in our study. Note that while IT2media provides their services to many clients, it is always their workers choice whether they participate in a study or not. So at the time the workers saw the sentence you referred to, they were already willing to participate in our study.
>
> Lastly, there might be a misunderstanding about the term "research project". Obviously, on the original website there was the German term "Forschungsprojekt". That term usually refers to publicly funded research projects (i.e. with a grant). For example, our project is funded by the European Comission and the state of North-Rhine-Westphalia. Publication of reproducible, scientific results is an integral and inherent part of research projects in the German-speaking community. To make people again aware of this, we have put that sentence there.
>
> > On page 26 it is stated that commenters "knowing that their texts might be subject to academic use and republishing." - while the current intended CC-BY-NC-SA license prohibits commerical use, it is not strictly limited to academic/research use.
>
> You are absolutely right to point out that the CC-BY-NC-SA license provides more degrees of freedom than just "academic use" or "republishing". Our primary intention was to share our enthusiasm to provide a large dataset under a permissive license. However, we understand that such statements might confuse any future users about the exact possible usages of the dataset.
>
> Consequently, we removed this statement. As we already pointed out above, we have all required legal permission on publishing and redistributing the dataset without limitations. See also our answer to  "Were the individuals in question notified about the data collection" in the datasheet. Here, we again would like to refer to the previously mentioned document about the permissions granted by Rheinische Post, which is available on [OpenReview.net](https://openreview.net/attachment?id=NfTU-wN8Uo&name=supplementary_material). Please note that not providing the data for commercial purposes is a decision we have made voluntarily and at our own request.
>
> We hope that this will clarify that the default CC-BY-NCA-SA 4.0 license applies; the information regarding the affected peoples' notification is restricted to the corresponding paragraphs.

---

> ### Author Response · Authors · 2021-09-28
> **Author Response IX**
>
> > Also it is not obvious that the comment authors have the IP rights on what they posted. In particular flagged comments could well include copyrighted material (which is problematic). E.g., what if one would copy parts of Harry Potter as comments? Such contents could be contained in this data set. People that post abusive content are likely not following IP rights well either.
>
> You are right that this could be a potential issue. However, all comments in our dataset are pre-moderated by employees of Rheinische Post, see section 3.1 of our paper. Therefore, if there was a comment including copyrighted material, the Rheinische Post would have deleted it from their database beforehand. As discussed, our dataset ranges from Nov 2018 to Jun 2020, which means that we received the data with a delay of about six month. This is quite a long timespan in which an IP holder could have notified Rheinische Post to request removal of potentially copyrighted material.
>
> Further, we also checked the comments regarding IP by looking at comments which were extra ordinarily long. This relates to the German copyright law ("Urheberrecht"), which requires artifacts to have a minimum level of creativity and contribution ("Schöpfungshöhe") to be subject for copyright at all. While this can obviously not be measured in an amount of characters it is much harder for short comments to fulfil this criteria.
>
> However, within the datasheet we specifically state our e-mail adresses for the case that comments could still contain passages which violate IP. Should this situation arise, we will of course delete the affected comment from the dataset.
>
> > "Does the dataset relate to people?" - while you do not include the identities of the commenters, the comments might relate to people. After all, some of these comments are insulting and otherwise abusive. Some of the comments themselves might be legally problematic to publish! What if there were a comment "<First Author> is a <insult> and lives at <home address>"? I do not think comments can easily be republished.
>
> Thank you for pointing this out. After reviewing our paper and the Datasheet we agree that further elaboration is required.
>     All published comments have already been checked by the community managers of our partner (the RP) for eligibility of being published online. Additional checks where necessary for those comments which have been withhold in the moderation process, as these might indeed contain problematic personal information (be it addresses, clear names, ...). Therefore, we checked each rejected comment to see if it contained any personal information before submitting it to the crowdworking platform. Critical instances were removed so that personal information does neither leak to the crowdworkers nor to the general public through the publication of this dataset.
>     To avoid confusion and doubts on the side of future readers, we also added the corresponding explanations to our paper and the Datasheet for datasets.
> > In A.4.1, there are eight people named that were "involved" "throughout the creation of the dataset". Of these, three were not made authors of the data set article, why are they not even credited? And the name that is not "involved", what is his contribution to the data set article?
>
> Thank you for pointing this out. All mentioned persons only assisted in few of the stages of the dataset creation and did not co-author the paper. It was important to us to acknowledge their contribution to the dataset itself, so we made sure to mention them in the accompanying dataset sheet. However, given the increased space for the paper revision, we will of course also acknowledge the three in the paper's acknowledgements.
>     The person that was not involved in the dataset creation is Moritz Vincent Seiler, who greatly assisted us with the experiments conducted based on the dataset. Furthermore, he helped in co-authoring the corresonding paper sections.
>
> > Additional Feedback:
> The authors should reconsider the placement of Figure 3 in between of two paragraphs, neither of which relates to Figure 3.
>
> As requested, we realigned Figure 3.
>
> > For the format of this submission, I would have expected more detailed documentation of the data labeling, and less focus on the preliminary experiments (which also are not a strength of the paper...)
>
> Utilising the additional page we address this issue by extending the description on the dataset creation process on page 4 f. We hope that via these changes your main concerns are addressed and that the value of the paper improved.

---

> ### Comment · Reviewer_y8tm · 2021-09-29
> **Thank the authors for the detailed updates**
>
> Thank you for the detailed updates. Some problems (in paritcular, ethical/legal issues) appear to have been cleared up, and hence I expect I will be able to increase my rating at least a bit. But I will likely need to re-read the updated paper, and I will not be able to do this within the discussion period.
> I assume that I remain skeptical that the data quality is good enough for this to be ultimately useful. But to a large extent, this will be up to the future users to decide. It will likely be used in some papers (as everything is used with x variants of deep neural networks these days), and hence will influence some research, but I do not expect this data set to ultimately bring research much forward. It will likely not enable deeper understanding of the methods, nor particularly powerful abusive language filtering (as it does not appear to transfer to other data; it may age quickly as political topics shift), but most likely will be just another data set to be included in benchmarks now and then. My main concern is that the experimental results are not too promising (results from RP do not transfer to the older derStandard) and that the chosen evaluation measures are too detached from the problem to solve.

---

### Official Review · Reviewer_bA2z · 2021-09-19
**Promising new dataset not without flaws**

**Rating:** 6
**Confidence:** 3
**Correctness:** Sound, but see metric comment above.
**Clarity:** Very well written and nice to read.

**Strengths:**

- Well motivated paper and useful dataset, with two separate profiles of classifying offensive comments.
- Dataset construction and analysis appears sound, with sensible labelling schema, annotation strategy, and quality assurance. The demographic overview is highly appreciated and differences in the annotation profiles are insightful.
- Large number of baselines were explored, in three different evaluation settings, and experimental comparison to the DerStandard dataset was appreciated, highlighting the benefits of the proposed one.

**Weaknesses:**

- As the authors pointed out, the IAA is very low with 0.19 and while they explored dataset versions with different agreement thresholds, according to Figure 2, the agreement remains very low. It would have been nice if the authors compared their agreement with relevant literature, putting the numbers better into perspective for this specific task.
- It is not entirely clear to me: the labels assigned by the Mods are also multiclass, but the authors do not compare these with the ones they have chosen for the crowdsource annotators. More insights would be appreciated.
- Further insights on the text form of the comments would be nice: are they colloquial with emojis, similarly to tweets and the likes or use standard German.
- I find 40% of the paper to be for the experiments a bit much, considering it is a dataset paper, I'd rather get more insights on the dataset specifics in the main paper, instead of the appendix. For instance, Figure 4 can be moved to the appendix and the joint task objective can be described more concisely.
- On that note, I would be curious to know whether the authors have considered using more task-relevant auxiliary objectives, e.g. sentiment classification (of tweets).
- The authors should motivate better their use of the employed metric: are we really interested in the entire spectrum of false negatives/positives? In many real-life applications, we probably want to minimize the number of abusive comments that are not flagged.

**Additional Feedback:**

l.135 quality insurance -> quality assurance

**Documentation:**

Carefully constructed and documented dataset, well documented, with filled-out Datasheet. As mentioned above, more focus on the dataset creation and less on the experiments in the main paper.

**Ethics:**

Dataset is made available with CC BY-NC-SA license, which is very nice considering the restrictions such datasets normally face.

**Relation To Prior Work:**

NA.

**Summary And Contributions:**

The authors present a novel abusive comment dataset for German, consisting of article's comments of the German newspaper Rheinische Post (RP), labelled by both moderators (Mod) and crowdworkers (Crowd), resulting in a dataset of 85000 annotated comments. Five crowd workers labeled each comment using a labeling schema of 7 classes, with several quality asssurance methods put in place, such as pre-screenings and continuous control by interleaving gold annotations. The paper analyses dataset characteristics such as comment rejection tendency based on the annotator demographics, showing they largely align with literature, but also highlighting differences between Mod and Crowd annotations, the latter rejecting comments more often. The dataset is evaluated in a binary setting (abusive, non-abusive) on several baselines and a more sophisticated line of transformer models, and compared to the most related dataset (DerStandard), indicating that the larger size of the proposed dataset helps to generalize to unseen domains, as a classifier solely trained on this dataset performs much better when tested on DerStandard than vice versa.

---

> ### Author Response · Authors · 2021-09-28
> **Author Response I**
>
> Thank you very much for your positive review, which we greatly appreciate! In the following we will adress your concerns individually in order. Additionally you find an updated version of our paper with changes highlighted in red.
>
> > Weaknesses:
> As the authors pointed out, the IAA is very low with 0.19 and while they explored dataset versions with different agreement thresholds, according to Figure 2, the agreement remains very low. It would have been nice if the authors compared their agreement with relevant literature, putting the numbers better into perspective for this specific task.
>
>
> We agree that the IAA for the complete dataset is quite low with a Krippendorffs $\alpha$ of 0.19. In the domain of abusive language detection this is however frequently observed and stated as an important challenge:
>
>     Vidgen, B., Harris, A., Nguyen, D., Tromble, R., Hale, S., & Margetts, H. (2019, August). Challenges and frontiers in abusive content detection. In Proceedings of the third workshop on abusive language online (pp. 80-93).
>
> One notable example is the study by Ross et al., who also analyzed German content (Tweets) and reported an agreement between 0.19 and 0.29 (depending on different groups):
>
>     Roß, B., Rist, M., Carbonell, G., Cabrera, B., Kurowsky, N., Wojatzki, M., 2016. Measuring the Reliability of Hate Speech Annotations: the Case of the European Refugee Crisis. In: Proceedings of NLP4CMC III: 3rd Workshop on Natural Language Processing for Computer-Mediated Communication
>
> The corresponding references and explanation have also been added to Section 4 to better contextualize our own IAA results.
>
> > It is not entirely clear to me: the labels assigned by the Mods are also multiclass, but the authors do not compare these with the ones they have chosen for the crowdsource annotators. More insights would be appreciated.
>
> Thank you also for pointing out the explanatory labeling imprecision, which we would like to clarify: The data we got from our partner, the Rheinische Post, is only **binary** coded (non-abusive [0] or abusive [1]). Their moderation system further allows the moderators to add additional (free-text) comments to their moderation decision in order to further explain their decisions. However, they use this feature irregularly (*only for a small subset of comments*) and inconsistently (*different moderators use different phrases (content, style, length)*). Hence, we have no usable additional label information, so after carefully reviewing our phrasing in the paper, we agree that it might cause confusion and rephrased it accordingly.
>
> > Further insights on the text form of the comments would be nice: are they colloquial with emojis, similarly to tweets and the likes or use standard German.
>
>
> Thank you for this suggestion which is something which should be investigated in-depth in the future. For now, we had an initial look at the data by investigating emoji and hashtag frequencies: out of the 85k comments, only 68 contain hashtags and 1,127 contain emojis. This is extremely low in comparison to other, for example, tweet-based datasets. In e.g.,
>
>     Ljubešić, N., & Fišer, D. (2016, August). A global analysis of emoji usage. In Proceedings of the 10th Web as Corpus Workshop (pp. 82-89).
>
> the authors report that 19.6% of the collected tweets contain emojis while in our dataset they are only present in 1.3% of all comments.
>
> The Rheinische Post does not enforce length restrictions (in contrast to Twitter) on their comments. Many answers tend to be long and often contain complete sentences (the average length of comments is 227 characters).
> Furthermore, our dataset differs from tweets in the fact that most commentators write entire sentences (on average 227 characters per comment).
> Based on our manual inspection of the dataset, the comments are mostly written in informal German – but this question may be answered by further studies in greater detail. The difference might further be explained by the system utilised by the commenters.

---

> ### Author Response · Authors · 2021-09-28
> **Author Response II**
>
>
> > I find 40% of the paper to be for the experiments a bit much, considering it is a dataset paper, I'd rather get more insights on the dataset specifics in the main paper, instead of the appendix. For instance, Figure 4 can be moved to the appendix and the joint task objective can be described more concisely.
>
>
> Thank you. For us, it is hard to find a perfect trade-off between dataset description and experiments. We agree that it is one of the central aspects of the paper to publish the dataset and thus needs a thorough introduction. However, we also believe that it is quite important for our readers to have an idea of what to expect from the dataset from an ML point of view. We used the space gained by the removal of Figure 4 (which we agree should be considered as further information for interested readers), as well as by getting the 10th page in the revision, to provide further details on the facilitation of the annotation study. Furthermore, we copied the table with the dataset structure from the Datasheet to the main paper, to better illustrate the data provided in our dataset.
>
>
>
> > On that note, I would be curious to know whether the authors have considered using more task-relevant auxiliary objectives, e.g. sentiment classification (of tweets).
>
>
> Yes, we did consider the utilisation of more task-relevant auxiliary objectives and agree that it should be mentioned in the paper as a future research endeavour. In fact, there already exist some promising approaches in this direction, e.g., in:
>
>     Rajamanikam, S., Mishra, P., Yannakoudakis, H., & Shutova, E. (2020). Joint modelling of emotion and abusive language detection. Assoc. Comput. Linguist.(ACL).
>
> Our dataset could be used to evaluate the generalisability of those approaches for this different domain of comments (and not only Tweet data). However, we think this is an interesting future research direction that is not in the scope of this initial dataset publication.
>
> > The authors should motivate better their use of the employed metric: are we really interested in the entire spectrum of false negatives/positives? In many real-life applications, we probably want to minimize the number of abusive comments that are not flagged.
>
> Thank you for your remark. You are absolutely right with your comment that this aspect was not properly motivated in our submitted version. As a consequence, we included a whole new paragraph in Section 5.4, where we elaborate on the importance of errors in the context of different scenarios. In terms of real-life applications, two prominent examples come to mind. Some newspapers utilise pre-moderation (moderating comments before they are published) while others utilise post-moderation (moderating comments after they are published, e.g., when they are reported by other users). The main goal of trained models is to support the moderator during the moderation process. If we assume that comments that are not flagged by our model as being abusive are directly published and those that are flagged are assigned to a manual moderator inspection, the main goal should be to minimise false negatives (abusive comments that are flagged as being unproblematic), as those are directly published on the newspapers’ website. On the other hand, false positives are comments that are flagged as being abusive although they are not. These comments have to be inspected by moderators manually, resulting in more workload for them. However, ultimately this is more acceptable than publishing a lot of abusive language comments which will probably result in negative publicity. Our point is that for different scenarios different errors are more or less important. This is why we think it is essential to report P-R and AUC curves as they allow us to investigate different thresholds (of course extreme solutions are mostly useless systems, as pointed out by Reviewer 4).
>
>
> > Documentation:
> Carefully constructed and documented dataset, well documented, with filled-out Datasheet. As mentioned above, more focus on the dataset creation and less on the experiments in the main paper.
>
>
> Thank you very much for your positive comment. Utilising the additional page we address this issue by extending the description on the dataset creation process on page 4f. Moreover, as suggested, we moved Figure 4 to the Appendix.
>
>
> > Additional Feedback:
> l.135 quality insurance -> quality assurance
>
>
> Thank you for highlighting this typographical error. We fixed this in the current version of the paper.

---

### Official Review · Reviewer_apYG · 2021-09-20
**Important dataset, but the contribution is limited and incremental**

**Rating:** 6
**Confidence:** 3
**Clarity:** Yes, the paper is well written.

**Strengths:**

* This dataset can suffice the need for a German news comment dataset. Each comment in the dataset is annotated by both professional moderators and 5 crowd-sourcing workers. Multiple annotations allow for creating more accurate ground truth.

* The authors have elaborate on dataset characteristics and present many valuable findings related to rejection tendency.

* Extensive experiments show that the dataset is a very useful resource for training abusive comment detection models.

**Weaknesses:**

This dataset, of course, can be an important corpus for the abusive comment detection domain. However, the contribution is limited to this small domain. The classification task proposed in the paper is relatively standard.
I also have ethical concerns about the annotation collection procedure (See Ethics)


**Additional Feedback:**

I would suggest that the authors deleted page 14.

**Correctness:**

Yes, the dataset is constructed soundly and the evaluation methods and experiment design are appropriate and performed correctly.

**Documentation:**

Yes, sufficient detail is provided.

**Ethics:**

I see that some of the crowdworkers’ ages are below 20. It is unclear that if any minors participate in this study and if minors were at risk of exposure to adult and violent content during the study.

**Relation To Prior Work:**

Yes, it is clearly discussed how the paper differs from previous contributions.

**Summary And Contributions:**

This work introduced a German abusive language comment dataset. This dataset includes both professional moderators’ and crowd-based annotations. The author presented characteristics of the dataset and the demographic properties of crowd-study participants. Experiments show that this dataset can serve as training data for abusive language detection models.

---

> ### Author Response · Authors · 2021-09-28
> **Author Response**
>
> Thank you for acknowledging the strengths of our paper and your valuable inputs. In the following we would like to address your remarks individually. We also updated our paper and highlighted the changes in red.
>
> > Weaknesses:
> This dataset, of course, can be an important corpus for the abusive comment detection domain. However, the contribution is limited to this small domain. The classification task proposed in the paper is relatively standard. I also have ethical concerns about the annotation collection procedure (See Ethics)
>
>
> Thank you for your positive feedback. We also think that we provide a valuable dataset to our research domain which currently lacks datasets and despite the fact that the task itself is quite standard (classification), current research still struggles to develop well-performing models (which again can be attributed to the scarce dataset situation). We think that the detection of abusive language, in general, is something that becomes more and more important, especially in times of increasing user participation on social media platforms. We think it is an important research area that strives to create a safe user environment and ultimately facilitates democratic principles such as freedom of expression. We acknowledge your ethical concerns and address them below.
>
>
> > Ethics:
> I see that some of the crowdworkers’ ages are below 20. It is unclear that if any minors participate in this study and if minors were at risk of exposure to adult and violent content during the study.
>
>
> Thank you for raising this potential issue. We verified that no minors were involved in our crowd-study. Our service provider (Crowdguru/IT2media GmbH & Co. KG) needs workers to confirm they are adults. Please note that some of the crowdworkers are between 18 and 20, as the legal age in Germany is 18. Moreover, we specifically made the crowdworkers aware of the fact that they would be confronted with abusive content before their participation in the study ("informed consent").
>
> > Additional Feedback:
> I would suggest that the authors deleted page 14.
>
>
> We agree and removed page 14 in our current revision.

---

### Official Review · Reviewer_phJr · 2021-09-23
**The authors publish and analyse the largest annotated 10 German abusive language comment datasets**

**Rating:** 6
**Confidence:** 5
**Clarity:** Yes

**Strengths:**

Largest dataset in German Language so far for abusive comment
Experimental evaluation includes traditional and state-of-the-art models

**Weaknesses:**

Contents from a single newspaper outlet. Also the commenters have to comment on the partner’s website. This also largely creates an inherent population bias in the dataset.
labeling schema is applied directly from another paper [36]
Even if extensive quality check is placed, the Krippendorff’s alpha [27] for all of the 177 85,000 comments we find low agreement (α = 0.19).

**Additional Feedback:**

Nice paper.

**Correctness:**

Yes. The dataset construction process is described nicely. The evaluation setup may  be improved further.

**Documentation:**

Yes. I cannot find URL to the dataset

**Relation To Prior Work:**

Yes

**Summary And Contributions:**

The authors publish and analyse the largest annotated 10 German abusive language comment datasets. They also conduct a thorough crowd-based annotation study that complements professional moderators’ decisions.

---

> ### Author Response · Authors · 2021-09-28
> **Author Response**
>
>
>
> Thank you very much for your positive feedback! We appreciate all of your comments and in the following, we would like to address your concerns in the order they were raised. Moreover you find an updated revision of our submission with updates highlighted in red.
>
> > Weaknesses:
> Contents from a single newspaper outlet. Also the commenters have to comment on the partner’s website. This also largely creates an inherent population bias in the dataset.
>
>
> We agree that it is not optimal to have only data from one specific newspaper outlet. However, as depicted in Table 1 of the presented paper, the majority of published newspaper comment datasets only contain comments from an individual outlet.
>     As described in the “Related Work” section, it is hard to find a comparable dataset in this domain because the newspapers are quite reluctant to publish their comment data. They have to change their terms of service and ensure that laws and regulations are not violated. Fortunately, in the context of our project, the Rheinische Post actually accounted for all of these aspects allowing us to publish the presented dataset. We think due to the data scarcity in the domain it is still a much-needed contribution. Furthermore, we highlight that most of the existing annotated datasets originate from Twitter and not from newspaper comment sections. Therefore, the mentioned potential population bias could be an opportunity to  investigate differences between this domain and traditional social media platforms such as Twitter or Facebook.
> > Even if extensive quality check is placed, the Krippendorff’s alpha [27] for all of the 177 85,000 comments we find low agreement (α = 0.19).
>
> Thank you for acknowleging our extensive quality checks. In the following, we would like to address your comment on the low Krippendorff’s $\alpha$ values: we highlight that these low agreement scores are frequently observed in the domain of abusive language detection, e.g., in
>
>     Wulczyn, E., Thain, N., and Dixon, L. 2017. “Ex Machina: Personal Attacks Seen at Scale,” in Proceedings of the 26th International Conference on World Wide Web, WWW ’17, Perth, Australia: ACM Press, pp. 1391–1399. (https://doi.org/10.1145/3038912.3052591).
>
> In the context of German datasets, there are nt many publications available due to the lack of datasets that exist for that language in general. In one notable study by Roß et al. who investigated on German hate-speech Twitter comments during the refugee crisis, an inter-annotator agreement between .18 to .29 was reported:
>
>     Roß, B., Rist, M., Carbonell, G., Cabrera, B., Kurowsky, N., Wojatzki, M., 2016. Measuring the Reliability of Hate Speech Annotations: the Case of the European Refugee Crisis. In: Proceedings of NLP4CMC III: 3rd Workshop on Natural Language Processing for Computer-Mediated Communication
>
> These low agreement scores can have multiple underlying reasons. Most importantly, there is no real upon agreed definition of what constitutes abusive language. Abusive language is a vague term, which is not necessarily a problem. As pointed out by Vidgen and Derczynski:
>
> *"However, this is not necessarily a ‘problem’ as much a research opportunity; abusive content annotation is better understood, ontologically, as an intersubjective process in which agreement is constructed, rather than an objective process in which a ‘true’ annotation is ‘found’. Because of the inherently subjective, contextual and political nature of abuse, some researchers have shifted the question of ‘how can we achieve the correct annotation?’ to ‘who should decide what the correct annotation is?’"* (Vidgen B, Derczynski L (2020) Directions in abusive language training data, a systematic review: Garbage in, garbage out. PLoS ONE 15(12): e0243300. https://doi.org/10.1371/journal.pone.0243300)
>
> We want to highlight that despite the low inter-annotator agreement on the complete dataset, it is possible to adjust this agreement by setting different thresholds. While during this process, we lose observations that are considered as being “controversial”, we increase the inter-annotator agreement. In the end, we leave this decision up to the researchers who work with our dataset as they can decide which threshold to apply. In some cases (depending on the research question at hand), it might be beneficial to investigate the comments with the low agreement explicitly.
>
> > Documentation:
> I cannot find URL to the dataset
>
>
> The URL (Sciebo link) to the dataset was provided via the corresponding section on OpenReview.net, which should only be visible to reviewers. In case this should not be visible to you, please contact the NEURIPS chairs.

---

### Author Response · Authors · 2021-09-29
**Thank you**

Dear Reviewers,

on behalf of my co-authors, we thank you again for your time and effort, reviewing our paper. As you see, we tried to answer all of your open questions in the comment section and hope that you are satisfied with our new revision, where we incorporated your feedback. We think, that your suggestions significantly improved our manuscript. We would highly appreciate feedback, whether you are satisfied with our answers and changes. If you still have further questions, do not hesitate to reach back, before the rebuttal phase closes. If we convinced you with our answers and changes, kindly reconsider updating your scores.

Kind regards,

Dennis Assenmacher

---

### Decision · Program_Chairs · 2021-10-11

**Decision:**

Accept

**Comment:**

This work constructs the largest annotated 10 German abusive language comment datasets and performs a comprehensive analysis. Although reviewers raise their concerns from different aspects, the authors have made convincing replies. I recommend it is accepted, and authors should continue to refine the draft based on the rebuttal comments.